

# Probabilistic programming in Python using PyMC3

John Salvatier[1], Thomas V. Wiecki[2] and Christopher Fonnesbeck[3]

[1] AI Impacts, Berkeley, CA, United States
[2] Quantopian Inc, Boston, MA, United States
[3] Department of Biostatistics, Vanderbilt University, Nashville, TN, United States

## ABSTRACT

Probabilistic programming allows for automatic Bayesian inference on user-defined probabilistic models. Recent advances in Markov chain Monte Carlo (MCMC) sampling allow inference on increasingly complex models. This class of MCMC, known as Hamiltonian Monte Carlo, requires gradient information which is often not readily available. PyMC3 is a new open source probabilistic programming framework written in Python that uses Theano to compute gradients via automatic differentiation as well as compile probabilistic programs on-the-fly to C for increased speed. Contrary to other probabilistic programming languages, PyMC3 allows model specification directly in Python code. The lack of a domain specific language allows for great flexibility and direct interaction with the model. This paper is a tutorial-style introduction to this software package.

## INTRODUCTION

Probabilistic programming (PP) allows for flexible specification and fitting of Bayesian statistical models. PyMC3 is a new, open-source PP framework with an intuitive and readable, yet powerful, syntax that is close to the natural syntax statisticians use to describe models. It features next-generation Markov chain Monte Carlo (MCMC) sampling algorithms such as the No-U-Turn Sampler (NUTS) (*Hoffman & Gelman, 2014*), a self-tuning variant of Hamiltonian Monte Carlo (HMC) (*Duane et al., 1987*). This class of samplers works well on high dimensional and complex posterior distributions and allows many complex models to be fit without specialized knowledge about fitting algorithms. HMC and NUTS take advantage of gradient information from the likelihood to achieve much faster convergence than traditional sampling methods, especially for larger models. NUTS also has several self-tuning strategies for adaptively setting the tuneable parameters of Hamiltonian Monte Carlo, which means specialized knowledge about how the algorithms work is not required. PyMC3, Stan (*Stan Development Team, 2015*), and the LaplacesDemon package for R are currently the only PP packages to offer HMC.

A number of probabilistic programming languages and systems have emerged over the past 2–3 decades. One of the earliest to enjoy widespread usage was the BUGS language (*Spiegelhalter et al., 1995*), which allows for the easy specification of Bayesian

Corresponding author
Thomas V. Wiecki,
thomas.wiecki@gmail.com

models, and fitting them via Markov chain Monte Carlo methods. Newer, more expressive languages have allowed for the creation of factor graphs and probabilistic graphical models. Each of these systems are domain-specific languages built on top of existing low-level languages; notable examples include Church (*Goodman et al., 2012*) (derived from Scheme), Anglican (*Wood, Van de Meent & Mansinghka, 2014*) (integrated with Clojure and compiled with a Java Virtual Machine), Venture (*Mansinghka, Selsam & Perov, 2014*) (built from C++), Infer.NET (*Minka et al., 2010*) (built upon the .NET framework), Figaro (*Pfeffer, 2014*) (embedded into Scala), WebPPL (*Goodman & Stuhlmüller, 2014*) (embedded into JavaScript), Picture (*Kulkarni et al., 2015*) (embedded into Julia), and Quicksand (*Ritchie, 2014*) (embedded into Lua).

Probabilistic programming in Python (*Python Software Foundation, 2010*) confers a number of advantages including multi-platform compatibility, an expressive yet clean and readable syntax, easy integration with other scientific libraries, and extensibility via C, C++, Fortran or Cython (*Behnel et al., 2011*). These features make it straightforward to write and use custom statistical distributions, samplers and transformation functions, as required by Bayesian analysis. While most of PyMC3's user-facing features are written in pure Python, it leverages Theano (*Bergstra et al., 2010*; *Bastien et al., 2012*) to transparently transcode models to C and compile them to machine code, thereby boosting performance. Theano is a library that allows expressions to be defined using generalized vector data structures called *tensors*, which are tightly integrated with the popular NumPy (*Van der Walt, Colbert, Varoquaux, 2011*) `ndarray` data structure, and similarly allow for broadcasting and advanced indexing, just as NumPy arrays do. Theano also automatically optimizes the likelihood's computational graph for speed and provides simple GPU integration.

Here, we present a primer on the use of PyMC3 for solving general Bayesian statistical inference and prediction problems. We will first describe basic PyMC3 usage, including installation, data creation, model definition, model fitting and posterior analysis. We will then employ two case studies to illustrate how to define and fit more sophisticated models. Finally we will show how PyMC3 can be extended and discuss more advanced features, such as the Generalized Linear Models (GLM) subpackage, custom distributions, custom transformations and alternative storage backends.

## INSTALLATION

Running PyMC3 requires a working Python interpreter (*Python Software Foundation, 2010*), either version 2.7 (or more recent) or 3.4 (or more recent); we recommend that new users install version 3.4. A complete Python installation for Mac OSX, Linux and Windows can most easily be obtained by downloading and installing the free AnacondaPythonDistribution by ContinuumIO.

PyMC3 can be installed using 'pip':

```
pip install git+https://github.com/pymc-devs/pymc3
```

PyMC3 depends on several third-party Python packages which will be automatically installed when installing via pip. The four required dependencies are: Theano, NumPy,

SciPy, and `Matplotlib`. To take full advantage of PyMC3, the optional dependencies `Pandas` and `Patsy` should also be installed.

```
pip install patsy pandas
```

The source code for PyMC3 is hosted on GitHub at https://github.com/pymc-devs/pymc3 and is distributed under the liberal ApacheLicense2.0. On the GitHub site, users may also report bugs and other issues, as well as contribute code to the project, which we actively encourage. Comprehensive documentation is readily available at http://pymc-devs.github.io/pymc3/.

## A MOTIVATING EXAMPLE: LINEAR REGRESSION

To introduce model definition, fitting and posterior analysis, we first consider a simple Bayesian linear regression model with normal priors on the parameters. We are interested in predicting outcomes $Y$ as normally-distributed observations with an expected value $\mu$ that is a linear function of two predictor variables, $X_1$ and $X_2$.

$$Y \sim \mathcal{N}(\mu, \sigma^2)$$
$$\mu = \alpha + \beta_1 X_1 + \beta_2 X_2$$

where $\alpha$ is the intercept, and $\beta_i$ is the coefficient for covariate $X_i$, while $\sigma$ represents the observation or measurement error. We will apply zero-mean normal priors with variance of 10 to both regression coefficients, which corresponds to weak information regarding the true parameter values. Since variances must be positive, we will also choose a half-normal distribution (normal distribution bounded below at zero) as the prior for $\sigma$.

$$\alpha \sim \mathcal{N}(0, 10)$$
$$\beta_i \sim \mathcal{N}(0, 10)$$
$$\sigma \sim |\mathcal{N}(0, 1)|.$$

### Generating data

We can simulate some data from this model using NumPy's `random` module, and then use PyMC3 to try to recover the corresponding parameters. The following code implements this simulation, and the resulting data are shown in Fig. 1:

```python
import numpy as np
import matplotlib.pyplot as plt

# Intialize random number generator
np.random.seed(123)

# True parameter values
alpha, sigma = 1, 1
beta = [1, 2.5]
```

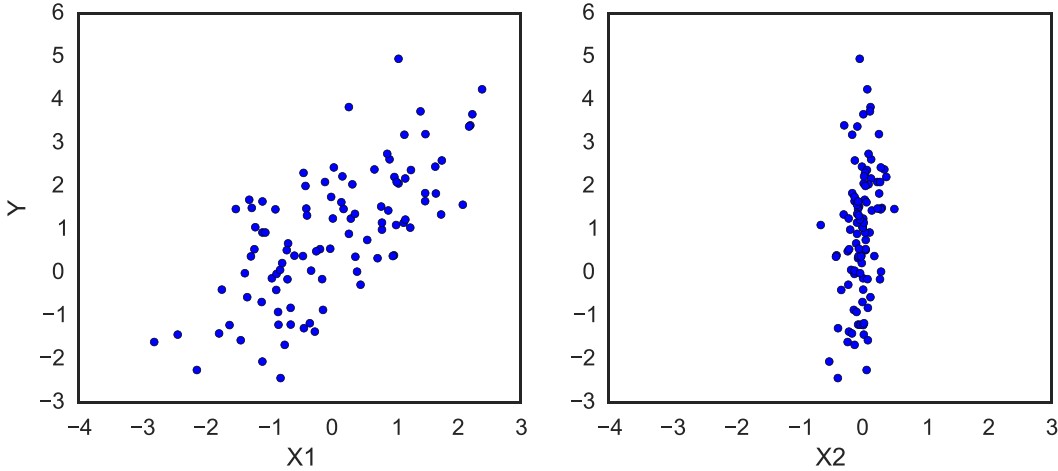

**Figure 1** Simulated regression data.

```
# Size of dataset
size = 100

# Predictor variable
X1 = np.linspace(0, 1, size)
X2 = np.linspace(0,.2, size)

# Simulate outcome variable
Y = alpha + beta[0]*X1 + beta[1]*X2 + np.random.randn(size)*sigma
```

## Model specification

Specifying this model in PyMC3 is straightforward because the syntax is similar to the statistical notation. For the most part, each line of Python code corresponds to a line in the model notation above. First, we import the components we will need from PyMC3.

```
from pymc3 import Model, Normal, HalfNormal
```

The following code implements the model in PyMC:

```
basic_model = Model()

with basic_model:

    # Priors for unknown model parameters
    alpha = Normal('alpha', mu=0, sd=10)
    beta = Normal('beta', mu=0, sd=10, shape=2)
    sigma = HalfNormal('sigma', sd=1)

    # Expected value of outcome
    mu = alpha + beta[0]*X1 + beta[1]*X2
```

```
# Likelihood (sampling distribution) of observations
Y_obs = Normal('Y_obs', mu=mu, sd=sigma, observed=Y)
```

The first line,

```
basic_model = Model()
```

creates a new `Model` object which is a container for the model random variables. Following instantiation of the model, the subsequent specification of the model components is performed inside a `with` statement:

```
with basic_model:
```

This creates a context manager, with our `basic_model` as the context, that includes all statements until the indented block ends. This means all PyMC3 objects introduced in the indented code block below the `with` statement are added to the model behind the scenes. Absent this context manager idiom, we would be forced to manually associate each of the variables with `basic_model` as they are created, which would result in more verbose code. If you try to create a new random variable outside of a model context manger, it will raise an error since there is no obvious model for the variable to be added to.

The first three statements in the context manager create **stochastic** random variables with Normal prior distributions for the regression coefficients, and a half-normal distribution for the standard deviation of the observations, $\sigma$.

```
alpha = Normal('alpha', mu=0, sd=10)
beta = Normal('beta', mu=0, sd=10, shape=2)
sigma = HalfNormal('sigma', sd=1)
```

These are stochastic because their values are partly determined by its parents in the dependency graph of random variables, which for priors are simple constants, and are partly random, according to the specified probability distribution.

The `Normal` constructor creates a normal random variable to use as a prior. The first argument for random variable constructors is always the *name* of the variable, which should almost always match the name of the Python variable being assigned to, since it can be used to retrieve the variable from the model when summarizing output. The remaining required arguments for a stochastic object are the parameters, which in the case of the normal distribution are the mean `mu` and the standard deviation `sd`, which we assign hyperparameter values for the model. In general, a distribution's parameters are values that determine the location, shape or scale of the random variable, depending on the parameterization of the distribution. Most commonly used distributions, such as `Beta`, `Exponential`, `Categorical`, `Gamma`, `Binomial` and others, are available as PyMC3 objects, and do not need to be manually coded by the user.

The `beta` variable has an additional `shape` argument to denote it as a vector-valued parameter of size 2. The `shape` argument is available for all distributions and specifies the length or shape of the random variable; when unspecified, it defaults to a value of one (i.e., a scalar). It can be an integer to specify an array, or a tuple to specify a multidimensional

array. For example, `shape=(5,7)` makes random variable that takes a 5 by 7 matrix as its value.

Detailed notes about distributions, sampling methods and other PyMC3 functions are available via the `help` function.

```
help(Normal)

Help on class Normal in module pymc3.distributions.continuous:

class Normal(pymc3.distributions.distribution.Continuous)
 |  Normal log-likelihood.
 |
 |  .. math::
ight\}
 |
 |  Parameters
 |  ----------
 |  mu : float
 |      Mean of the distribution.
 |  tau : float
 |      Precision of the distribution, which corresponds to
 |      :math:'1/\sigma^2' (tau > 0).
 |  sd : float
 |    Standard deviation of the distribution. Alternative parameterization.
 |
 |  .. note::
 |  - :math:'E(X) = \mu'
 |  - :math:'Var(X) = 1 / \tau'
```

Having defined the priors, the next statement creates the expected value `mu` of the outcomes, specifying the linear relationship:

```
mu = alpha + beta[0]*X1 + beta[1]*X2
```

This creates a **deterministic** random variable, which implies that its value is completely determined by its parents' values. That is, there is no uncertainty in the variable beyond that which is inherent in the parents' values. Here, `mu` is just the sum of the intercept `alpha` and the two products of the coefficients in `beta` and the predictor variables, whatever their current values may be.

PyMC3 random variables and data can be arbitrarily added, subtracted, divided, or multiplied together, as well as indexed (extracting a subset of values) to create new random variables. Many common mathematical functions like `sum`, `sin`, `exp` and linear algebra functions like `dot` (for inner product) and `inv` (for inverse) are also provided. Applying operators and functions to PyMC3 objects results in tremendous model expressivity.

The final line of the model defines `Y_obs`, the sampling distribution of the response data.

```
Y_obs = Normal('Y_obs', mu=mu, sd=sigma, observed=Y)
```

This is a special case of a stochastic variable that we call an **observed stochastic**, and it is the data likelihood of the model. It is identical to a standard stochastic, except that

its `observed` argument, which passes the data to the variable, indicates that the values for this variable were observed, and should not be changed by any fitting algorithm applied to the model. The data can be passed in the form of either a `numpy.ndarray` or `pandas.DataFrame` object.

Notice that, unlike the prior distributions, the parameters for the normal distribution of `Y_obs` are not fixed values, but rather are the deterministic object `mu` and the stochastic `sigma`. This creates parent-child relationships between the likelihood and these two variables, as part of the directed acyclic graph of the model.

## Model fitting

Having completely specified our model, the next step is to obtain posterior estimates for the unknown variables in the model. Ideally, we could derive the posterior estimates analytically, but for most non-trivial models this is not feasible. We will consider two approaches, whose appropriateness depends on the structure of the model and the goals of the analysis: finding the *maximum a posteriori* (MAP) point using optimization methods, and computing summaries based on samples drawn from the posterior distribution using MCMC sampling methods.

### *Maximum a posteriori methods*

The **maximum a posteriori (MAP)** estimate for a model, is the mode of the posterior distribution and is generally found using numerical optimization methods. This is often fast and easy to do, but only gives a point estimate for the parameters and can be misleading if the mode isn't representative of the distribution. PyMC3 provides this functionality with the `find_MAP` function.

Below we find the MAP for our original model. The MAP is returned as a parameter **point**, which is always represented by a Python dictionary of variable names to NumPy arrays of parameter values.

```
from pymc3 import find_MAP

map_estimate = find_MAP(model=basic_model)

print(map_estimate)

{'alpha': array(1.0136638069892534),
'beta': array([ 1.46791629,  0.29358326]),
'sigma_log': array(0.11928770010017063)}
```

By default, `find_MAP` uses the Broyden–Fletcher–Goldfarb–Shanno (BFGS) optimization algorithm to find the maximum of the log-posterior but also allows selection of other optimization algorithms from the `scipy.optimize` module. For example, below we use Powell's method to find the MAP.

```
from scipy import optimize
```

```
map_estimate = find_MAP(model=basic_model, fmin=optimize.fmin_powell)

print(map_estimate)

{'alpha': array(1.0175522109423465),
'beta': array([ 1.51426782,  0.03520891]),
'sigma_log': array(0.11815106849951475)}
```

It is important to note that the MAP estimate is not always reasonable, especially if the mode is at an extreme. This can be a subtle issue; with high dimensional posteriors, one can have areas of extremely high density but low total probability because the volume is very small. This will often occur in hierarchical models with the variance parameter for the random effect. If the individual group means are all the same, the posterior will have near infinite density if the scale parameter for the group means is almost zero, even though the probability of such a small scale parameter will be small since the group means must be extremely close together.

Also, most techniques for finding the MAP estimate only find a *local* optimium (which is often good enough), and can therefore fail badly for multimodal posteriors if the different modes are meaningfully different.

### Sampling methods

Though finding the MAP is a fast and easy way of obtaining parameter estimates of well-behaved models, it is limited because there is no associated estimate of uncertainty produced with the MAP estimates. Instead, a simulation-based approach such as MCMC can be used to obtain a Markov chain of values that, given the satisfaction of certain conditions, are indistinguishable from samples from the posterior distribution.

To conduct MCMC sampling to generate posterior samples in PyMC3, we specify a **step method** object that corresponds to a single iteration of a particular MCMC algorithm, such as Metropolis, Slice sampling, or the No-U-Turn Sampler (NUTS). PyMC3's `step_methods` submodule contains the following samplers: `NUTS`, `Metropolis`, `Slice`, `HamiltonianMC`, and `BinaryMetropolis`.

### Gradient-based sampling methods

PyMC3 implements several standard sampling algorithms, such as adaptive Metropolis-Hastings and adaptive slice sampling, but PyMC3's most capable step method is the No-U-Turn Sampler. NUTS is especially useful for sampling from models that have many continuous parameters, a situation where older MCMC algorithms work very slowly. It takes advantage of information about where regions of higher probability are, based on the gradient of the log posterior-density. This helps it achieve dramatically faster convergence on large problems than traditional sampling methods achieve. PyMC3 relies on Theano to analytically compute model gradients via automatic differentiation of the posterior density. NUTS also has several self-tuning strategies for adaptively setting the tunable parameters of Hamiltonian Monte Carlo. For random variables that are undifferentiable (namely, discrete variables) NUTS cannot be used, but it may still be used on the differentiable variables in a model that contains undifferentiable variables.

NUTS requires a scaling matrix parameter, which is analogous to the variance parameter for the jump proposal distribution in Metropolis-Hastings, although NUTS uses it somewhat differently. The matrix gives an approximate shape of the posterior distribution, so that NUTS does not make jumps that are too large in some directions and too small in other directions. It is important to set this scaling parameter to a reasonable value to facilitate efficient sampling. This is especially true for models that have many unobserved stochastic random variables or models with highly non-normal posterior distributions. Poor scaling parameters will slow down NUTS significantly, sometimes almost stopping it completely. A reasonable starting point for sampling can also be important for efficient sampling, but not as often.

Fortunately, NUTS can often make good guesses for the scaling parameters. If you pass a point in parameter space (as a dictionary of variable names to parameter values, the same format as returned by `find_MAP`) to NUTS, it will look at the local curvature of the log posterior-density (the diagonal of the Hessian matrix) at that point to guess values for a good scaling vector, which can result in a good value. The MAP estimate is often a good point to use to initiate sampling. It is also possible to supply your own vector or scaling matrix to NUTS. Additionally, the `find_hessian` or `find_hessian_diag` functions can be used to modify a Hessian at a specific point to be used as the scaling matrix or vector.

Here, we will use NUTS to sample 2000 draws from the posterior using the MAP as the starting and scaling point. Sampling must also be performed inside the context of the model.

```python
from pymc3 import NUTS, sample

with basic_model:

    # obtain starting values via MAP
    start = find_MAP(fmin=optimize.fmin_powell)

    # instantiate sampler
    step = NUTS(scaling=start)

    # draw 2000 posterior samples
    trace = sample(2000, step, start=start)
```

```
 [----------------100%-----------------] 2000 of 2000 complete in 4.6 sec
```

The `sample` function runs the step method(s) passed to it for the given number of iterations and returns a `Trace` object containing the samples collected, in the order they were collected. The `trace` object can be queried in a similar way to a `dict` containing a map from variable names to `numpy.arrays`. The first dimension of the array is the sampling index and the later dimensions match the shape of the variable. We can extract the last 5 values for the `alpha` variable as follows

```python
trace['alpha'][-5:]
```

```
array([ 0.98134501,  1.04901676,  1.03638451,  0.88261935,  0.95910723])
```

## Posterior analysis

PyMC3 provides plotting and summarization functions for inspecting the sampling output. A simple posterior plot can be created using `traceplot`, its output is shown in Fig. 2.

```
from pymc3 import traceplot

traceplot(trace)
```

The left column consists of a smoothed histogram (using kernel density estimation) of the marginal posteriors of each stochastic random variable while the right column contains the samples of the Markov chain plotted in sequential order. The `beta` variable, being vector-valued, produces two histograms and two sample traces, corresponding to both predictor coefficients.

For a tabular summary, the `summary` function provides a text-based output of common posterior statistics:

```
from pymc3 import summary

summary(trace['alpha'])

alpha:

  Mean             SD               MC Error         95% HPD interval
  -------------------------------------------------------------------

  1.024            0.244            0.007            [0.489, 1.457]

  Posterior quantiles:
  2.5              25               50               75               97.5
  |--------------|==============|==============|--------------|

  0.523            0.865            1.024            1.200            1.501
```

## CASE STUDY 1: STOCHASTIC VOLATILITY

We present a case study of stochastic volatility, time varying stock market volatility, to illustrate PyMC3's capability for addressing more realistic problems. The distribution of market returns is highly non-normal, which makes sampling the volatilities significantly more difficult. This example has 400+ parameters so using older sampling algorithms like Metropolis-Hastings would be inefficient, generating highly auto-correlated samples with a low effective sample size. Instead, we use NUTS, which is dramatically more efficient.

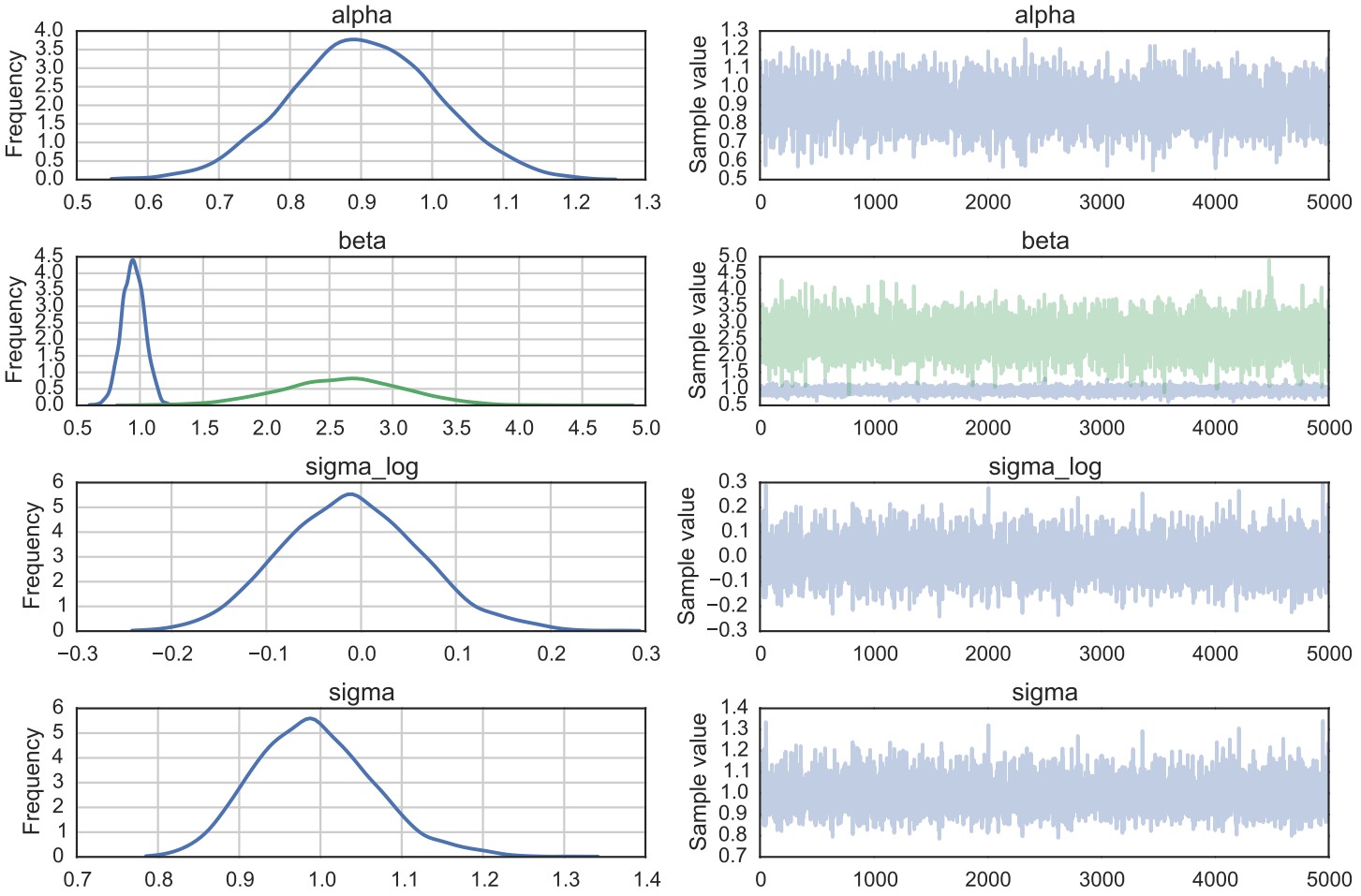

**Figure 2** Kernel density estimates and simulated trace for each variable in the linear regression model.

## The model

Asset prices have time-varying volatility (variance of day over day `returns`). In some periods, returns are highly variable, while in others they are very stable. Stochastic volatility models address this with a latent volatility variable, which is allowed to change over time. The following model is similar to the one described in the NUTS paper (*Hoffman & Gelman, 2014*, p. 21).

$$\sigma \sim \exp(50)$$
$$\nu \sim \exp(.1)$$
$$s_i \sim \mathcal{N}(s_{i-1}, \sigma^{-2})$$
$$\log(y_i) \sim T(\nu, 0, \exp(-2s_i)).$$

Here, $y$ is the response variable, a daily return series which we model with a Student-T distribution having an unknown degrees of freedom parameter, and a scale parameter determined by a latent process $s$. The individual $s_i$ are the individual daily log volatilities in the latent log volatility process.

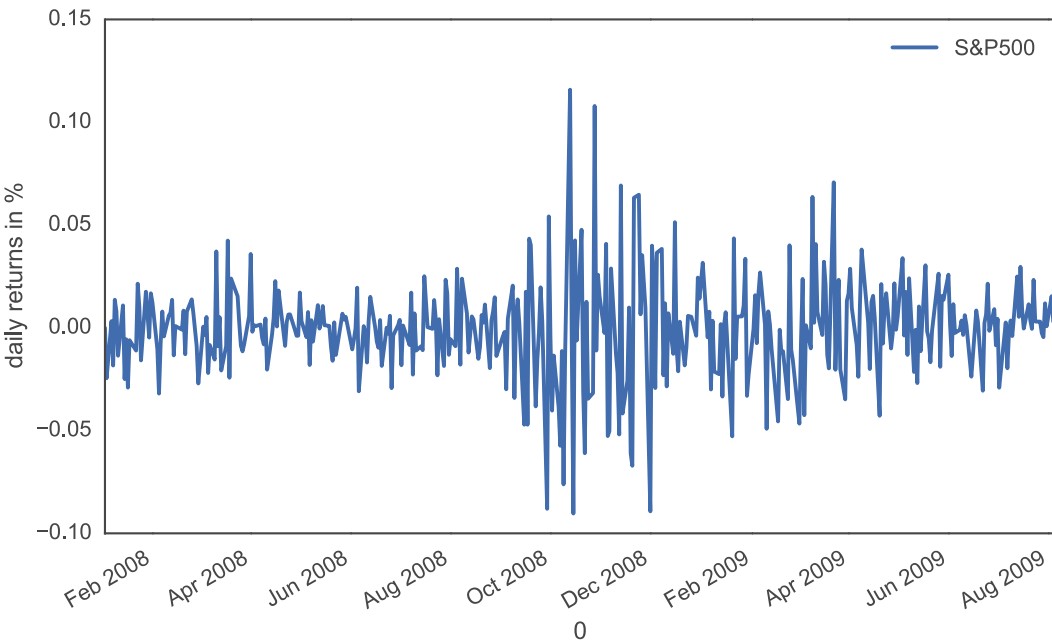

**Figure 3** **Historical daily returns of the S&P500 during the 2008 financial crisis.**

## The data

Our data consist of daily returns of the S&P 500 during the 2008 financial crisis.

```
import pandas as pd
returns = pd.read_csv('data/SP500.csv', index_col=0, parse_dates=True)
```

See Fig. 3 for a plot of the daily returns data. As can be seen, stock market volatility increased remarkably during the 2008 financial crisis.

## Model implementation

As with the linear regression example, implementing the model in PyMC3 mirrors its statistical specification. This model employs several new distributions: the `Exponential` distribution for the $\nu$ and $\sigma$ priors, the Student-T (`StudentT`) distribution for distribution of returns, and the `GaussianRandomWalk` for the prior for the latent volatilities.

In PyMC3, variables with positive support like `Exponential` are transformed with a log transform, making sampling more robust. Behind the scenes, the variable is transformed to the unconstrained space (named "variableName_log") and added to the model for sampling. In this model this happens behind the scenes for both the degrees of freedom, `nu`, and the scale parameter for the volatility process, `sigma`, since they both have exponential priors. Variables with priors that are constrained on both sides, like `Beta` or `Uniform`, are also transformed to be unconstrained, here with a log odds transform.

Although (unlike model specification in PyMC2) we do not typically provide starting points for variables at the model specification stage, it is possible to provide an initial value for any distribution (called a "test value" in Theano) using the `testval` argument. This overrides the default test value for the distribution (usually the mean, median or mode of

the distribution), and is most often useful if some values are invalid and we want to ensure we select a valid one. The test values for the distributions are also used as a starting point for sampling and optimization by default, though this is easily overriden.

The vector of latent volatilities s is given a prior distribution by a `GaussianRandomWalk` object. As its name suggests, GaussianRandomWalk is a vector-valued distribution where the values of the vector form a random normal walk of length n, as specified by the `shape` argument. The scale of the innovations of the random walk, `sigma`, is specified in terms of the precision of the normally distributed innovations and can be a scalar or vector.

```
from pymc3 import Exponential, StudentT, exp, Deterministic
from pymc3.distributions.timeseries import GaussianRandomWalk

with Model() as sp500_model:

    nu = Exponential('nu', 1./10, testval=5.)

    sigma = Exponential('sigma', 1./.02, testval=.1)

    s = GaussianRandomWalk('s', sigma**-2, shape=len(returns))

    volatility_process = Deterministic('volatility_process', exp(-2*s))

    r = StudentT('r', nu, lam=1/volatility_process, observed=returns['S&P500'])
```

Notice that we transform the log volatility process s into the volatility process by `exp(-2*s)`. Here, `exp` is a Theano function, rather than the corresponding function in NumPy; Theano provides a large subset of the mathematical functions that NumPy does.

Also note that we have declared the `Model` name `sp500_model` in the first occurrence of the context manager, rather than splitting it into two lines, as we did for the first example.

### Fitting

Before we draw samples from the posterior, it is prudent to find a decent starting value, by which we mean a point of relatively high probability. For this model, the full *maximum a posteriori* (MAP) point over all variables is degenerate and has infinite density. But, if we fix `log_sigma` and `nu` it is no longer degenerate, so we find the MAP with respect only to the volatility process s keeping `log_sigma` and `nu` constant at their default values (remember that we set `testval=.1` for `sigma`). We use the Limited-memory BFGS (L-BFGS) optimizer, which is provided by the `scipy.optimize` package, as it is more efficient for high dimensional functions; this model includes 400 stochastic random variables (mostly from s).

As a sampling strategy, we execute a short initial run to locate a volume of high probability, then start again at the new starting point to obtain a sample that can be used for inference. `trace[-1]` gives us the last point in the sampling trace. NUTS will recalculate the scaling parameters based on the new point, and in this case it leads to faster sampling due to better scaling.

```
import scipy
with sp500_model:
```

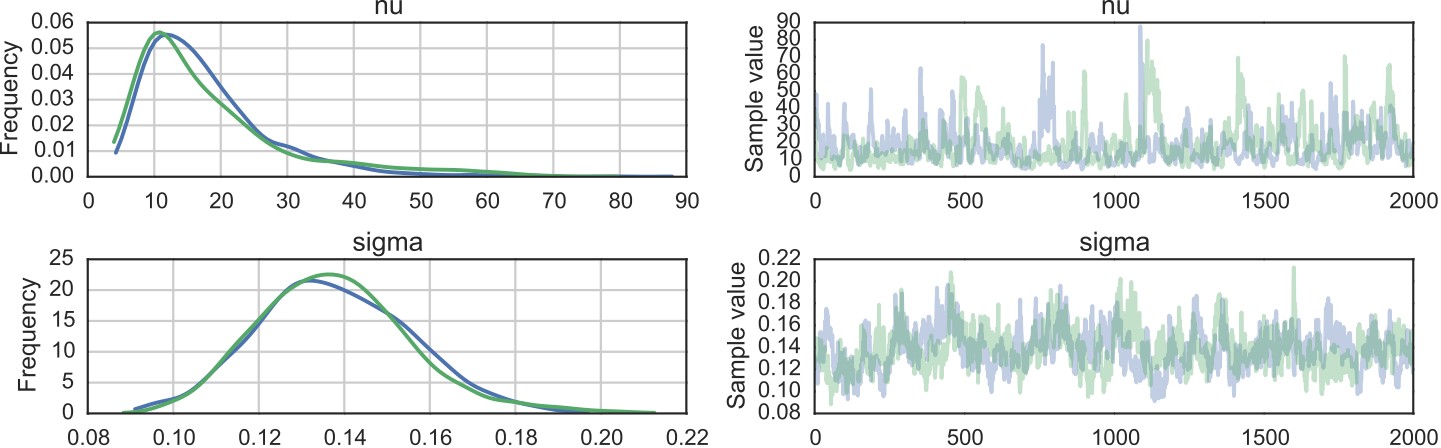

**Figure 4** **Posterior samples of degrees of freedom** (nu) **and scale** (`sigma`) **parameters of the stochastic volatility model.** Each plotted line represents a single independent chain sampled in parallel.

```
start = find_MAP(vars=[s], fmin=scipy.optimize.fmin_l_bfgs_b)

step = NUTS(scaling=start)
trace = sample(100, step, progressbar=False)

# Start next run at the last sampled position.
step = NUTS(scaling=trace[-1], gamma=.25)
trace = sample(2000, step, start=trace[-1], progressbar=False, njobs=2)
```

Notice that the call to `sample` includes an optional `njobs=2` argument, which enables the parallel sampling of 4 chains (assuming that we have 2 processors available).

We can check our samples by looking at the traceplot for nu and `sigma`; each parallel chain will be plotted within the same set of axes (Fig. 4).

```
traceplot(trace, [nu, sigma]);
```

Finally we plot the distribution of volatility paths by plotting many of our sampled volatility paths on the same graph (Fig. 5). Each is rendered partially transparent (via the `alpha` argument in Matplotlib's `plot` function) so the regions where many paths overlap are shaded more darkly.

```
fig, ax = plt.subplots(figsize=(15, 8))
returns.plot(ax=ax)
ax.plot(returns.index, 1/np.exp(trace['s',::30].T), 'r', alpha=.03);
ax.set(title='volatility_process', xlabel='time', ylabel='volatility');
ax.legend(['S&P500', 'stochastic volatility process'])
```

As you can see, the model correctly infers the increase in volatility during the 2008 financial crash.

It is worth emphasizing the complexity of this model due to its high dimensionality and dependency-structure in the random walk distribution. NUTS as implemented in PyMC3, however, correctly infers the posterior distribution with ease.

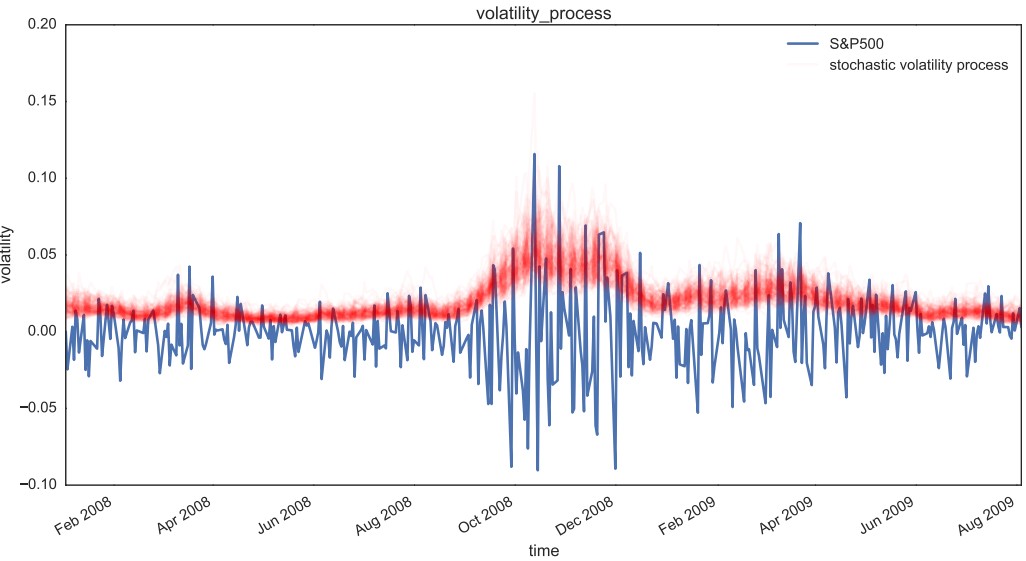

**Figure 5** Posterior plot of volatility paths (red), alongside market data (blue).

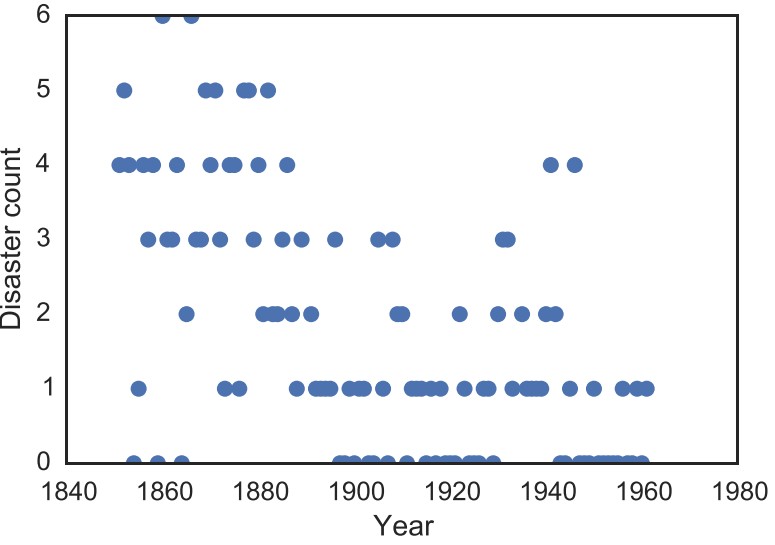

**Figure 6** Recorded counts of coal mining disasters in the UK, 1851–1962.

## CASE STUDY 2: COAL MINING DISASTERS

This case study implements a change-point model for a time series of recorded coal mining disasters in the UK from 1851 to 1962 (*Jarrett, 1979*). The annual number of disasters is thought to have been affected by changes in safety regulations during this period, as can be seen in Fig. 6. We have also included a pair of years with missing data, identified as missing by a NumPy `MaskedArray` using -999 as a sentinel value.

Our objective is to estimate when the change occurred, in the presence of missing data, using multiple step methods to allow us to fit a model that includes both discrete and continuous random variables.

```
disaster_data = np.ma.masked_values([4, 5, 4, 0, 1, 4, 3, 4, 0, 6, 3, 3, 4, 0, 2, 6,
                                     3, 3, 5, 4, 5, 3, 1, 4, 4, 1, 5, 5, 3, 4, 2, 5,
                                     2, 2, 3, 4, 2, 1, 3, -999, 2, 1, 1, 1, 1, 3, 0, 0,
                                     1, 0, 1, 1, 0, 0, 3, 1, 0, 3, 2, 2, 0, 1, 1, 1,
                                     0, 1, 0, 1, 0, 0, 0, 2, 1, 0, 0, 0, 1, 1, 0, 2,
                                     3, 3, 1, -999, 2, 1, 1, 1, 1, 2, 4, 2, 0, 0, 1, 4,
                                     0, 0, 0, 1, 0, 0, 0, 0, 0, 1, 0, 0, 1, 0, 1], value=-999)
year = np.arange(1851, 1962)

plot(year, disaster_data, 'o', markersize=8);
ylabel("Disaster count")
xlabel("Year")
```

Counts of disasters in the time series is thought to follow a Poisson process, with a relatively large rate parameter in the early part of the time series, and a smaller rate in the later part. The Bayesian approach to such a problem is to treat the change point as an unknown quantity in the model, and assign it a prior distribution, which we update to a posterior using the evidence in the dataset.

In our model,

$$D_t \sim \text{Pois}(r_t)$$

$$r_t = \begin{cases} l, & \text{if } t < s \\ e, & \text{if } t \geq s \end{cases}$$

$$s \sim \text{Unif}(t_l, t_h)$$

$$e \sim \exp(1)$$

$$l \sim \exp(1)$$

the parameters are defined as follows:

- $D_t$: The number of disasters in year $t$
- $r_t$: The rate parameter of the Poisson distribution of disasters in year $t$.
- $s$: The year in which the rate parameter changes (the switchpoint).
- $e$: The rate parameter before the switchpoint $s$.
- $l$: The rate parameter after the switchpoint $s$.
- $t_l$, $t_h$: The lower and upper boundaries of year $t$.

```
from pymc3 import DiscreteUniform, Poisson, switch

with Model() as disaster_model:

    switchpoint = DiscreteUniform('switchpoint', lower=year.min(),
                        upper=year.max(), testval=1900)

    # Priors for pre- and post-switch rates number of disasters
    early_rate = Exponential('early_rate', 1)
```

```
            late_rate = Exponential('late_rate', 1)

            # Allocate appropriate Poisson rates to years before and after current
            rate = switch(switchpoint >= year, early_rate, late_rate)

            disasters = Poisson('disasters', rate, observed=disaster_data)
```

This model introduces discrete variables with the Poisson likelihood and a discrete-uniform prior on the change-point s. Our implementation of the `rate` variable is as a conditional deterministic variable, where its value is conditioned on the current value of s.

```
rate = switch(switchpoint >= year, early_rate, late_rate)
```

The conditional statement is realized using the Theano function `switch`, which uses the first argument to select either of the next two arguments.

Missing values are handled concisely by passing a `MaskedArray` or a `pandas.DataFrame` with NaN values to the `observed` argument when creating an observed stochastic random variable. From this, PyMC3 automatically creates another random variable, `disasters.missing_values`, which treats the missing values as unobserved `stochastic` nodes. All we need to do to handle the missing values is ensure we assign a step method to this random variable.

Unfortunately, because they are discrete variables and thus have no meaningful gradient, we cannot use NUTS for sampling either `switchpoint` or the missing disaster observations. Instead, we will sample using a `Metroplis` step method, which implements self-tuning Metropolis-Hastings, because it is designed to handle discrete values.

Here, the `sample` function receives a list containing both the NUTS and `Metropolis` samplers, and sampling proceeds by first applying `step1` then `step2` at each iteration.

```
from pymc3 import Metropolis

with disaster_model:
    step1 = NUTS([early_rate, late_rate])

    step2 = Metropolis([switchpoint, disasters.missing_values[0]] )

    trace = sample(10000, step=[step1, step2])

[----------------100%----------------] 10000 of 10000 complete in 6.9 sec
```

In the trace plot (Fig. 7) we can see that there is about a 10 year span that's plausible for a significant change in safety, but a 5-year span that contains most of the probability mass. The distribution is jagged because of the jumpy relationship between the year switch-point and the likelihood and not due to sampling error.

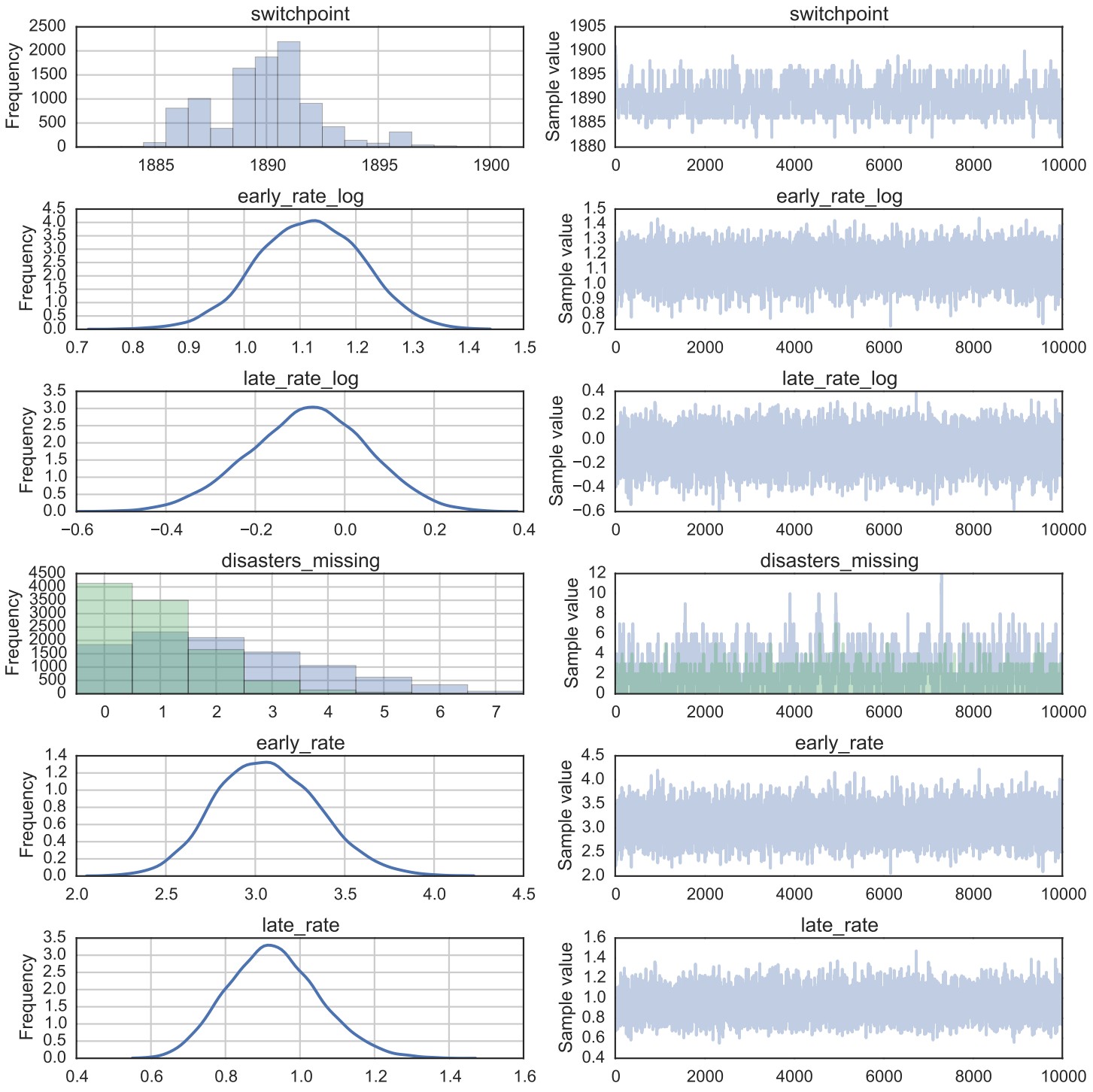

**Figure 7** Posterior distributions and traces from disasters change point model.

## PYMC3 FEATURES

### Arbitrary deterministic variables

Due to its reliance on Theano, PyMC3 provides many mathematical functions and operators for transforming random variables into new random variables. However, the library of functions in Theano is not exhaustive, therefore PyMC3 provides functionality for creating arbitrary Theano functions in pure Python, and including these functions in PyMC3 models. This is supported with the `as_op` function decorator.

```python
import theano.tensor as T
from theano.compile.ops import as_op

@as_op(itypes=[T.lscalar], otypes=[T.lscalar])
def crazy_modulo3(value):
    if value > 0:
        return value % 3
    else :
        return (-value + 1) % 3

with Model() as model_deterministic:
    a = Poisson('a', 1)
    b = crazy_modulo3(a)
```

Theano requires the types of the inputs and outputs of a function to be declared, which are specified for `as_op` by `itypes` for inputs and `otypes` for outputs. An important drawback of this approach is that it is not possible for Theano to inspect these functions in order to compute the gradient required for the Hamiltonian-based samplers. Therefore, it is not possible to use the `HMC` or `NUTS` samplers for a model that uses such an operator. However, it is possible to add a gradient if we inherit from `theano.Op` instead of using `as_op`.

### Arbitrary distributions

The library of statistical distributions in PyMC3, though large, is not exhaustive, but PyMC allows for the creation of user-defined probability distributions. For simple statistical distributions, the `DensityDist` function takes as an argument any function that calculates a log-probability $\log(p(x))$. This function may employ other parent random variables in its calculation. Here is an example inspired by a blog post by *VanderPlas (2014)*, where Jeffreys priors are used to specify priors that are invariant to transformation. In the case of simple linear regression, these are:

$$\beta \propto (1 + \beta^2)^{3/2}$$
$$\sigma \propto \frac{1}{\sigma}.$$

The logarithms of these functions can be specified as the argument to `DensityDist` and inserted into the model.

```
import theano.tensor as T
from pymc3 import DensityDist, Uniform

with Model() as model:
    alpha = Uniform('intercept', -100, 100)

    # Create custom densities
    beta = DensityDist('beta', lambda value: -1.5 * T.log(1 + value**2), testval=0)
    eps = DensityDist('eps', lambda value: -T.log(T.abs_(value)), testval=1)

    # Create likelihood
    like = Normal('y_est', mu=alpha + beta * X, sd=eps, observed=Y)
```

For more complex distributions, one can create a subclass of `Continuous` or `Discrete` and provide the custom `logp` function, as required. This is how the built-in distributions in PyMC3 are specified. As an example, fields like psychology and astrophysics have complex likelihood functions for a particular process that may require numerical approximation. In these cases, it is impossible to write the function in terms of predefined Theano operators and we must use a custom Theano operator using `as_op` or inheriting from `theano.Op`.

Implementing the `beta` variable above as a `Continuous` subclass is shown below, along with a sub-function using the `as_op` decorator, though this is not strictly necessary.

```
from pymc3.distributions import Continuous

class Beta(Continuous):
    def __init__(self, mu, *args, **kwargs):
        super(Beta, self).__init__(*args, **kwargs)
        self.mu = mu
        self.mode = mu

    def logp(self, value):
        mu = self.mu
        return beta_logp(value - mu)

@as_op(itypes=[T.dscalar], otypes=[T.dscalar])
def beta_logp(value):
    return -1.5 * np.log(1 + (value)**2)

with Model() as model:
    beta = Beta('slope', mu=0, testval=0)
```

## Generalized linear models

The generalized linear model (GLM) is a class of flexible models that is widely used to estimate regression relationships between a single outcome variable and one or multiple predictors. Because these models are so common, PyMC3 offers a `glm` submodule that

allows flexible creation of simple GLMs with an intuitive R-like syntax that is implemented via the `patsy` module.

The `glm` submodule requires data to be included as a `pandas DataFrame`. Hence, for our linear regression example:

```
# Convert X and Y to a pandas DataFrame
import pandas
df = pandas.DataFrame({'x1': X1, 'x2': X2, 'y': Y})
```

The model can then be very concisely specified in one line of code.

```
from pymc3.glm import glm

with Model() as model_glm:
    glm('y ~ x1 + x2', df)
```

The error distribution, if not specified via the `family` argument, is assumed to be normal. In the case of logistic regression, this can be modified by passing in a `Binomial` family object.

```
from pymc3.glm.families import Binomial

df_logistic = pandas.DataFrame({'x1': X1, 'x2': X2, 'y': Y > 0})

with Model() as model_glm_logistic:
    glm('y ~ x1 + x2', df_logistic, family=Binomial())
```

Models specified via `glm` can be sampled using the same `sample` function as standard PyMC3 models.

### Backends

PyMC3 has support for different ways to store samples from MCMC simulation, called backends. These include storing output in-memory, in text files, or in a SQLite database. By default, an in-memory `ndarray` is used but for very large models run for a long time, this can exceed the available RAM, and cause failure. Specifying a SQLite backend, for example, as the `trace` argument to `sample` will instead result in samples being saved to a database that is initialized automatically by the model.

```
from pymc3.backends import SQLite

with model_glm_logistic:
    backend = SQLite('logistic_trace.sqlite')
    trace = sample(5000, Metropolis(), trace=backend)

 [----------------100%-----------------] 5000 of 5000 complete in 2.0 sec
```

A secondary advantage to using an on-disk backend is the portability of model output, as the stored trace can then later (e.g., in another session) be re-loaded using the `load` function:

```
from pymc3.backends.sqlite import load

with basic_model:
    trace_loaded = load('logistic_trace.sqlite')
```

## DISCUSSION

Probabilistic programming is an emerging paradigm in statistical learning, of which Bayesian modeling is an important sub-discipline. The signature characteristics of probabilistic programming–specifying variables as probability distributions and conditioning variables on other variables and on observations–makes it a powerful tool for building models in a variety of settings, and over a range of model complexity. Accompanying the rise of probabilistic programming has been a burst of innovation in fitting methods for Bayesian models that represent notable improvement over existing MCMC methods. Yet, despite this expansion, there are few software packages available that have kept pace with the methodological innovation, and still fewer that allow non-expert users to implement models.

PyMC3 provides a probabilistic programming platform for quantitative researchers to implement statistical models flexibly and succinctly. A large library of statistical distributions and several pre-defined fitting algorithms allows users to focus on the scientific problem at hand, rather than the implementation details of Bayesian modeling. The choice of Python as a development language, rather than a domain-specific language, means that PyMC3 users are able to work interactively to build models, introspect model objects, and debug or profile their work, using a dynamic, high-level programming language that is easy to learn. The modular, object-oriented design of PyMC3 means that adding new fitting algorithms or other features is straightforward. In addition, PyMC3 comes with several features not found in most other packages, most notably Hamiltonian-based samplers as well as automatical transforms of constrained random variables which is only offered by STAN. Unlike STAN, however, PyMC3 supports discrete variables as well as non-gradient based sampling algorithms like Metropolis-Hastings and Slice sampling.

Development of PyMC3 is an ongoing effort and several features are planned for future versions. Most notably, variational inference techniques are often more efficient than MCMC sampling, at the cost of generalizability. More recently, however, black-box variational inference algorithms have been developed, such as automatic differentiation variational inference (ADVI) (*Kucukelbir et al., 2015*). This algorithm is slated for addition to PyMC3. As an open-source scientific computing toolkit, we encourage researchers developing new fitting algorithms for Bayesian models to provide reference implementations in PyMC3. Since samplers can be written in pure Python code, they can be implemented generally to make them work on arbitrary PyMC3 models, giving authors a larger audience to put their methods into use.

### Funding

The authors received no funding for this work.

## Competing Interests

Thomas V. Wiecki is an employee of Quantopian Inc. John Salvatier is an employee of AI Impacts.

## Author Contributions

- John Salvatier, Thomas V. Wiecki and Christopher Fonnesbeck conceived and designed the experiments, performed the experiments, analyzed the data, wrote the paper, prepared figures and/or tables, performed the computation work, reviewed drafts of the paper.

## Data Availability

https://github.com/pymc-devs/uq_chapter.

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
