# Peer review of "Probabilistic programming in Python using PyMC3"

_PeerJ Computer Science, doi:10.7717/peerj-cs.55_

## Round 0.1 · original submission · Major Revisions

We have received two reviews, which are enough to make a decision, since the reviews are consistent. The publisher has provided the opinion that "this article is in scope and (assuming it is correct etc) should be published." Hence the authors may ignore the (scope related) statements to the contrary in the reviews. However, please address all the specific issues raised in the reviews. We will likely send the revised version back to the first reviewer (not the second) and also to at least one other reviewer.

·

Basic reporting

This article is well-written. It is clear, understandable, and
self-contained. The prose is tight; the examples useful, and the
figures tidy.

I only found one error, which is on page 14: r_t should equal "e" if
t>=s, not "r".

Experimental design

This article should be considered documentation for a software
package. It is not "scientific", in the sense that there is no
clearly defined research question, at least not which is made explicit
in the text. Rather than closing a "knowledge gap", as described in
the editorial review standards, this software closes a "practice gap".

The implicit aim of probabilistic programming is to cleanly separate
model-building and inference, and make probabilistic modeling more
accessible to a broader audience by reducing implementation
complexity. This software package does indeed make strides towards
that goal, and does so with clarity and high standards.

It is not clear if software documentation is within the scope of PeerJ
or not.

Validity of the findings

There are no data, conclusions or experiments. Since this is software
documentation, the paper mostly consists of details of how to install
it, and examples of how to use it.

Very few implementation details are given, although the high level
bits are included (object-oriented interface to constructing models;
Theano for automatic differentiation; NUTS+HMC for inference on
continuous variables; basic MCMC for sampling discrete variables).

The paper does not really contain enough information for an individual
to replicate it, but it's not clear that "reproducibility" is the
correct rubric for this sort of work.

Additional comments

In my opinion, the PyMC3 package should not be considered a "probabilistic programming" language; rather, it should be considered an API for constructing graphical models.

I draw a clear distinction between the two; I reserve the phrase "probabilistic programming" for systems which are "turing-complete", in the sense that they can model nonparametric distributions, recursive distributions, programs that can write programs, higher-order functions, and the inclusion of arbitrary (stateless) deterministic functions in the middle of probabilistic models. As far as I can tell, PyMC3 does not support any of these.

PyMC3 seems much more comparable to, say, BUGS or BNT than to, say, Church or IBAL. I would therefore not call it "probabilistic programming" at all.

I would strongly encourage the authors to change the title and introduction to reflect this, to help keep the terminology consistent throughout the community.

Reviewer 2 ·

Basic reporting

Key prior work is not mentioned or contextualized. For example, Figaro is a Scala-embedded probabilistic programming system that exposes customizable inference strategies; WebPPL is another embedded language (embedded into JavaScript); Picture is another embedded language (into Julia); and Quicksand is another (into Lua).

Figure 6 is of very low resolution and needs to be redone.

Experimental design

As I understand the scope of the journal, the article may be suitable for publication as a PeerJ Preprint (after significant revision), but it does not describe original primary research. There is no research question or knowledge gap that is identified; instead, the paper is (clearly stated to be) a tutorial about a software system.

Validity of the findings

There are no conclusions, only capability demonstrations. The capability demonstrations would be stronger if some data about runtime performance and accuracy were included (including variability across ~10+ replications).

Additional comments

I want to encourage the authors to continue their work developing PyMC3 as a software tool, and to explore the question of what research questions in probabilistic programming they might be able to best study from the vantage point of their platform.

---

## Round 0.2 · accepted · Accept

Thank you for addressing the previous reviewers' comments. We have not succeeded in getting re-reviews, but the changes are small enough that acceptance now is warranted. Small note: please add citations for Figaro, WebPPL, Picture, and Quicksand.